# Evaluation of a pharmacist-led actionable audit and feedback intervention for improving medication safety in UK primary care: An interrupted time series analysis

Niels Peek[1,2,3]*, Wouter T. Gude[4], Richard N. Keers[1,5,6], Richard Williams[1,3], Evangelos Kontopantelis[7], Mark Jeffries[1,5], Denham L. Phipps[1,5], Benjamin Brown[1,3,8], Anthony J. Avery[1,9], Darren M. Ashcroft[1,2,5,7]

1 NIHR Greater Manchester Patient Safety Translational Research Centre, University of Manchester, Manchester Academic Health Science Centre, Manchester, United Kingdom, 2 NIHR Manchester Biomedical Research Centre, University of Manchester, Manchester Academic Health Science Centre, Manchester, United Kingdom, 3 Centre for Health Informatics, Division of Informatics, Imaging and Data Science, School of Health Sciences, Faculty of Biology, Medicine and Health, University of Manchester, Manchester Academic Health Science Centre, Manchester, United Kingdom, 4 Department of Medical Informatics, Amsterdam UMC, Amsterdam Public Health Research Institute, University of Amsterdam, Amsterdam, Netherlands, 5 Centre for Pharmacoepidemiology and Drug Safety, Division of Pharmacy and Optometry, School of Health Sciences, Faculty of Biology, Medicine and Health, University of Manchester, Manchester Academic Health Sciences Centre, Manchester, United Kingdom, 6 Pharmacy Department, Greater Manchester Mental Health NHS Foundation Trust, Manchester, United Kingdom, 7 NIHR School for Primary Care Research, University of Manchester, Manchester Academic Health Science Centre, Manchester, United Kingdom, 8 Centre for Primary Care, Division of Population Health, Health Services Research and Primary Care, School of Health Sciences, Faculty of Biology, Medicine and Health, University of Manchester, Manchester Academic Health Science Centre, Manchester, United Kingdom, 9 Division of Primary Care, School of Medicine, University of Nottingham, Nottingham, United Kingdom

* niels.peek@manchester.ac.uk

## Abstract

### Background

We evaluated the impact of the pharmacist-led Safety Medication dASHboard (SMASH) intervention on medication safety in primary care.

### Methods and findings

SMASH comprised (1) training of clinical pharmacists to deliver the intervention; (2) a web-based dashboard providing actionable, patient-level feedback; and (3) pharmacists reviewing individual at-risk patients, and initiating remedial actions or advising general practitioners on doing so. It was implemented in 43 general practices covering a population of 235,595 people in Salford (Greater Manchester), UK. All practices started receiving the intervention between 18 April 2016 and 26 September 2017. We used an interrupted time series analysis of rates (prevalence) of potentially hazardous prescribing and inadequate blood-test monitoring, comparing observed rates post-intervention to extrapolations from a 24-month pre-intervention trend. The number of people registered to participating practices and having 1 or more risk factors for being exposed to hazardous prescribing or inadequate blood-test

**Data Availability Statement:** The patient-level data for this study cannot be shared publicly because of

their sensitive nature (patient-level healthcare data, collected routinely without consent). They were provided for use in this study only by the Salford Integrated Record (SIR) Board. A copy can be requested from the SIR Board via SALCCG. SIRProposals@nhs.net using reference number DATA002. We have made the aggregated data, derived from the patient-level data, available through Mendeley Data at https://doi.org/10. 17632/ps8jwmmnkv.1. These are the weekly counts, per participating practice, for numerator and denominator, for each safety indicator over the 3-year study period.

**Funding:** The National Institute for Health Research (www.nihr.ac.uk) Greater Manchester Patient Safety Translational Research Centre (NIHR Greater Manchester PSTRC; PSTRC-2016-003) supported the time and facilities of NP, RNK, RW, MJ, DLP, AJA, and DMA. MRC (mrc.ukri.org) Health eResearch Centre grant MR/K006665/1 supported the time and facilities of EK. The funders had no role in study design, data collection and analysis, decision to publish, or preparation of the manuscript.

**Competing interests:** I have read the journal's policy and the authors of this manuscript have the following competing interests: RNK has received an honorarium for presenting on the topic of schizophrenia (epidemiology, aetiology, presentation and management) at the MORPh Consultancy Ltd Primary Care Pharmacists Training Network Mental Health Event, on 23rd October 2019, Liverpool UK.

**Abbreviations:** GP, general practitioner; ITS, interrupted time series; NHS, National Health Service; NOAC, non–vitamin K antagonist oral anticoagulant; NSAID, nonsteroidal anti-inflammatory drug; SIR, Salford Integrated Record; SMASH, Safety Medication dASHboard.

monitoring at the start of the intervention was 47,413 (males: 23,073 [48.7%]; mean age: 60 years [standard deviation: 21]). At baseline, 95% of practices had rates of potentially hazardous prescribing (composite of 10 indicators) between 0.88% and 6.19%. The prevalence of potentially hazardous prescribing reduced by 27.9% (95% CI 20.3% to 36.8%, $p < 0.001$) at 24 weeks and by 40.7% (95% CI 29.1% to 54.2%, $p < 0.001$) at 12 months after introduction of SMASH. The rate of inadequate blood-test monitoring (composite of 2 indicators) reduced by 22.0% (95% CI 0.2% to 50.7%, $p = 0.046$) at 24 weeks; the change at 12 months (23.5%) was no longer significant (95% CI −4.5% to 61.6%, $p = 0.127$). After 12 months, 95% of practices had rates of potentially hazardous prescribing between 0.74% and 3.02%. Study limitations include the fact that practices were not randomised, and therefore unmeasured confounding may have influenced our findings.

## Conclusions

The SMASH intervention was associated with reduced rates of potentially hazardous prescribing and inadequate blood-test monitoring in general practices. This reduction was sustained over 12 months after the start of the intervention for prescribing but not for monitoring of medication. There was a marked reduction in the variation in rates of hazardous prescribing between practices.

## Author summary

### Why was this study done?

- Electronic, actionable audit and feedback interventions can reduce the rates of potentially hazardous prescribing and inadequate blood-test monitoring in general practice.

- When based on a single feedback cycle, the effect of these interventions tends to wane over time.

- Therefore, we evaluated whether such an intervention is associated with sustained changes when it provides continuous, patient-level feedback.

### What did the researchers do and find?

- We evaluated the impact of the Safety Medication dASHboard (SMASH), a pharmacist-led intervention, in primary care during a phased rollout in Salford, UK (population size, 251,300).

- We found that the intervention was associated with a sustained reduction in potentially hazardous prescribing, and a reduction in the variation between practices, until the study was completed (after 12 months).

- A marked reduction in inadequate blood-test monitoring was also seen initially, but this reduction was not sustained over time.

**What do these findings mean?**

- Electronic, actionable audit and feedback interventions can provide long-term improvements to medication safety in general practice when they are delivered on an ongoing basis.

- More research is needed to obtain sustained reductions in inadequate blood-test monitoring.

## Introduction

Improving medication safety is a core objective for healthcare systems worldwide, and was recently identified by the World Health Organization (WHO) as the theme for the Third Global Patient Safety Challenge [1]. Incidents related to drugs and other treatments account for the largest proportion of preventable patient harm [2], and it has been estimated that the cost associated with medication errors reaches US$42 billion annually across the globe [3]. While most medications are prescribed, dispensed, and administered in ambulatory care settings, research and quality improvement have traditionally focused on hospital-based settings. Yet there is clear evidence that medication errors are very common also in ambulatory care and contribute to iatrogenic morbidity [4], despite the fact that individual items carry low risk. The sheer volume of ambulatory care prescribing (over 1 billion prescription items supplied in the community in England each year) means that avoidable deaths in primary care due to medication errors are 7 times likely than in secondary care (627 versus 85) [5]. Stocks et al. [6] found that 5% of patients who are managed in general practices in the UK are exposed to potentially hazardous prescribing, and about 12% had no record of appropriate monitoring. Variation in the prevalence of potentially high-risk prescribing and lack of appropriate monitoring tests between practices was high, even after adjustment for patient- and practice-level variables.

Information technology (IT) has long been recognised as having the potential to increase patient safety [7]. When healthcare providers switched to electronic prescribing in the late 1990s, computerised decision support was identified as a technology that could prevent prescribing errors by issuing alerts during the prescribing process. All electronic prescribing systems used in English primary care include decision support. However, clinicians experience alerts in most cases as 'too much, too late' [8], and it has been reported that up to 80% of alerts in ambulatory settings may be overridden [9].

Electronic, actionable audit and feedback [10] is a different approach that informs clinicians when potential errors have happened, prompting them to take remedial action. It can be applied instead of computerised decision support, or provided as an additional 'defence layer' [11]. The PINCER trial demonstrated that a pharmacist-led intervention in which trained pharmacists worked collaboratively with general practitioners (GPs) to act upon computer-generated feedback that identified potentially at-risk patients reduced the rates of hazardous prescribing and inadequate monitoring of medication [12]. The intervention was shown to be more effective than feedback alone, indicating that pharmacist visits played a crucial role in solving medication safety errors. However, the effect of PINCER waned by 12 months [12], probably because pharmacist involvement ceased after 3 months and the intervention relied

on a single feedback cycle—while feedback is known to be more effective when it is provided more than once [13].

We developed the pharmacist-led Safety Medication dASHboard (SMASH) intervention, building on the principles of the PINCER intervention, in which trained clinical pharmacists work with general practices to act upon feedback that is provided in a continuous fashion. To do this, we developed a novel electronic, interactive medication safety dashboard that identifies patients exposed to potential medication safety hazards and is updated on a daily basis [14], and capitalised on the National Health Service (NHS) England's policy to increase clinical pharmacists working as part of general practice teams [15–17]. Our objective was to evaluate changes in the prevalence of hazardous (high-risk) prescribing and inadequate blood-test monitoring associated with this intervention.

## Methods

### Study design

Our study took place during a phased rollout of the intervention in Salford, UK (population size, 251,300). Due to the planned rollout, randomisation was not possible. General practices received the intervention at different points in time, depending on the availability of trained pharmacists to help deliver this intervention within the practices. Once a pharmacist was available, the practice controlled the precise date at which it would start the intervention. We used an interrupted time series (ITS) design to evaluate the impact of the intervention on medication safety in this 'natural experiment'. Our description follows the TREND statement [18] for improving the reporting quality of nonrandomised evaluations of interventions (see S1 TREND Statement) and the Template for Intervention Description and Replication (TIDieR) Checklist [19] (see S1 TIDieR Checklist).

### Safety indicators

We used 12 medication safety indicators (10 relating to potentially hazardous prescribing and 2 to inadequate blood-test monitoring) developed for PINCER [12,20]. These 12 indicators were selected from a broader set of 24 indicators based on their prevalence nationally and in Salford (for an overview, see Stocks et al. [6] and Akbarov et al. [21], respectively), leaving out indicators with a low prevalence. Each indicator consisted of a denominator and a numerator. The denominator included all patients having 1 or more risk factors for being exposed to potentially hazardous prescribing or inadequate monitoring because of an existing diagnosis or prescribing pattern (during a time period specific to the indicator definition and the audit date). The numerator consisted of those patients who actually received the potentially hazardous prescription or had no record of the required monitoring during the time period leading up to the audit date. Operational definitions for each medication safety indicator are provided in Table A in S1 Table.

### Participants and recruitment

General practices were eligible to participate in the study if they had access to the Salford Integrated Record (SIR)—a data warehouse containing linked data from primary care and secondary data that is refreshed on a nightly basis [22]. Practices were recruited through local quality networks. In the United Kingdom, patients, including those individuals who are living in care home settings, are registered with 1 general practice that holds the patient's electronic health record, including diagnoses, test results, and details of prescribed medication. Patients were included in the study if they were registered with one of the participating practices and had the

potential to trigger one of the indicators at any time point during the 12 months that the intervention was deployed in their practice.

## Intervention

SMASH is an enhanced PINCER intervention as defined by the National Institute for Health and Care Excellence (NICE) medicine optimisation guideline [23], and comprised 3 components. First, clinical pharmacists worked in general practices as members of the practice team. They were trained to deliver the intervention and apply root cause analysis techniques to identify, explore, resolve, and prevent medication errors in partnership with general practice staff [24]. Some pharmacists had existing roles in general practices, and others were recruited into similar roles for practices where no pharmacist was employed previously. Many pharmacists worked across several practices. Second, pharmacists and practice staff were given access to a web-based, interactive dashboard that provided feedback on potentially hazardous prescribing. The dashboard was designed in collaboration with various stakeholders (pharmacists, GPs, and a patient). It generated lists of patients currently exposed to potentially hazardous prescribing or inadequate blood-test monitoring as defined by the 12 indicators, thus providing actionable feedback at the patient level [14]. The dashboard also provided practice-level summary data (the number and percentage of patients who concurrently had risk factors for each indicator, comparisons with the average for each indicator across all practices, and patterns over time) as well as educational material (evidence summaries for each indicator, links to the relevant literature, and details of possible actions that could be taken for each indicator to reduce risk to patients). Dashboard content (numbers and graphs) for each indicator was updated on a daily basis (at midnight) with up-to-date data on new diagnoses, laboratory test results, and prescriptions from the SIR data warehouse. Figs A–D in S1 Fig provide screenshots of the dashboard. Third, pharmacists reviewed individual patients whose records triggered the indicators, and initiated remedial actions (e.g., ordering laboratory tests) or advised GPs on action plans (e.g., discontinuing medication, co-prescribing protective medication) [25,26]. Pharmacists and GPs had access to the dashboard for the entire 12-month study period, and pharmacists were encouraged to continue monitoring the practice's dashboard and take action when new cases of hazardous prescribing and inadequate monitoring emerged. A small financial reward (£2.56 [US$3.31] per eligible patient) was also provided by the local NHS body responsible for the planning and commissioning of healthcare services to practices implementing the intervention [27].

## Outcome measures

Our primary outcome measures were prevalence of exposure to (a) any potentially hazardous prescribing (10 indicators) and (b) any inadequate blood-test monitoring (2 indicators) among patients with risk factors for such prescribing and monitoring (as defined by individual indicators), respectively. These outcomes were assessed through 2 composite measures, defined as the sum of all patients who triggered any prescribing (or, separately, monitoring) indicator divided by the sum of all patients with risk factors at a given audit date. Secondary outcome measures included rates of ongoing (existent for 30 or more days) and new (arisen within the previous 30 days) exposure to any potentially hazardous prescribing and any inadequate blood-test monitoring, and the prevalence of the 12 indicators individually. In each practice, outcomes were assessed every 4 weeks during the 24 months before and the 12 months after the practice started the intervention. The patient cohort in which the outcomes were evaluated was dynamic (i.e., the individual patients and the numbers of patients included in the denominator could change from 1 audit date to the next).

All outcome data were collected by interrogating primary care electronic health records from the SIR data warehouse for relevant diagnoses, prescriptions, and laboratory test results. Patients were censored from the study when they died or when they were no longer registered at one of the participating general practices.

## Statistical analysis

On the basis of baseline levels of potentially hazardous prescribing across practices in Salford [6], we assumed a baseline risk of 6%, with the intervention reducing it to 4.5%, and estimated that our study would have 88.3% power to detect this reduction in 2,000 patients across 30 practices with a statistical significance of 0.05.

The analyses were performed according to the intention-to-treat principle; we analysed data from all the eligible patients regardless of whether they actually received a medication review. We used an ITS design, to account for non-flat pre-intervention outcome trends in study outcomes. This involved using autoregressive integrated moving average (ARIMA) models with various post-intervention periods (4 weeks, 12 weeks, 24 weeks, and 12 months), through the *itsa* command in Stata version 15 [28,29]. Since ARIMA models are linear, outcomes were quantified as percentages and transformed to the log scale prior to analyses. The rates of potentially hazardous prescribing and inadequate monitoring are subject to ceiling and floor effects, which necessarily limits any linear trend in improvement. Therefore, we conducted the analyses on measures transformed to a logit scale, which has no ceiling [30]. Results were back-transformed to percentages to facilitate interpretation.

To account for and quantify practice variability, data from each practice were analysed separately in an ITS model and then aggregated using a random-effects meta-analysis model. More specifically, for each practice we used the estimates that are reported as standard in an ITS analysis (pre-slope, level change, and slope change [31]) in linear post-estimation commands. Through these, we evaluated differences in the outcomes at specific time points post-intervention (observed), compared to model extrapolations from the pre-intervention trend assuming no intervention (expected), on the log scale. The time points were 4 weeks, 12 weeks, 24 weeks, and 12 months after practices started the intervention. Results for these time points were next meta-analysed using a bootstrap version of the DerSimonian–Laird variance estimator [32,33], implemented in the *metaan* command in Stata [34], which also allows for back-transformation to percentages. This process allowed us to display practice heterogeneity in the intervention with forest plots, and also quantify it with an estimate of the routinely used $I^2$ statistic [35]. All statistical analyses were pre-specified except for the post-intervention periods shorter than 12 months (4 weeks, 12 weeks, and 24 weeks) at which we assessed changes in outcomes.

## Ethical approval

The study was approved on behalf of the UK's Health Research Authority by the North West–Greater Manchester East Research Ethics Committee on 24 September 2015 (reference 15/NW/0792). Patient consent was deemed not necessary because all electronic health record data were deidentified before they were shared with the study team.

## Patient and public involvement

A member of our patient and public involvement group at the NIHR Greater Manchester Patient Safety Translational Research Centre and other key stakeholders, including 6 GPs and 7 pharmacists, were involved in the design of the dashboard in an iterative process of short interviews and prototype reviews [36]. We met regularly with members of our patient and

public involvement group to review progress during the rollout of the intervention and will continue to work with this group to advise on plans for dissemination of these findings.

## Results

### Participating practices

Of the 44 general practices in Salford, 43 (98%) participated in the study. The missing practice wanted to participate but was ineligible because it did not contribute to the SIR data warehouse. The first practice started receiving the intervention from 18 April 2016, and the last from 26 September 2017. Start dates were not uniformly distributed over time; there was a peak in practices starting the intervention in November–December 2016. The participating practices had a pooled list size of 235,595 patients on the date that the practices started the intervention. A total of 54,044 patients had the potential to be identified against any of the 12 indicators in the 24 months before the start of the intervention, and 53,068 patients had this potential in the 12 months of the study. All these patients were included in the analysis at 1 or more time points.

Table 1 shows the baseline characteristics of the participating practices divided into 3 groups based on their starting period. The number of people registered to practices and having 1 or more risk factors for any of the medication safety indicators at the start of the intervention was 47,413. Of those, 23,073 (48.7%) were male, their mean age was 60 years (standard deviation: 21), and 24,954 (52.6%) had been prescribed 5 or more drugs simultaneously at that point in time. Of the 47,413 patients, there were 1,291 (2.7%) who were then exposed to a form of potentially hazardous prescribing or inadequate blood-test monitoring as defined by the 12 indicators.

**Table 1. Baseline characteristics of the included patients and participating practices, according to their start date.**

| Characteristic | Intervention start date | | |
|---|---|---|---|
| | **18 Apr to 12 Sep 2016** | **31 Oct to 31 Dec 2016** | **20 Jan to 26 Sep 2017** |
| **Practices** | 11 | 22 | 10 |
| Registered patients; mean (range) | 8,093 (3,318–15,104) | 5,096 (970–11,517) | 3,447 (1,951–8,695) |
| QOF points achieved[†]; mean (range) | 539 (482–558) | 482 (321–558) | 511 (450–557) |
| **Patients with 1 or more risk factors at the start of the intervention[‡]** | 17,211 | 23,744 | 6,458 |
| Exposure to 1 or more medication safety hazards; *n* (%) | 470 (2.7) | 589 (2.5) | 232 (3.6) |
| Age in years; mean (SD) | 58 (21) | 62 (21) | 57 (21) |
| Male sex; *n* (%) | 8,569 (49.8) | 11,302 (47.6) | 3,202 (49.6) |
| Residence in most deprived quintile of Salford postal codes; *n* (%) | 6,738 (39.1) | 9,020 (38.0) | 3,169 (49.1) |
| Charlson comorbidity index [46] ≥ 5; *n* (%) | 4,403 (25.6) | 7,581 (31.9) | 1,683 (26.1) |
| Polypharmacy (≥5 drugs); *n* (%) | 8,516 (49.5) | 13,166 (55.4) | 3,272 (50.7) |

[†]The Quality and Outcomes Framework (QOF) financially rewards primary care practices according to their performance on a range of clinical and organisational indicators, each of which is associated with a number of maximum achievable points, with each point corresponding to a defined payment. Presented QOF scores are from 2015–2016 and ranged from 0 to 559, with higher scores indicating better performance [47].

[‡]Included patients were people registered to practices and having 1 or more risk factors for any of the 12 medication safety indicators at the start of the intervention. This is the denominator for the remaining rows in the table.

## Rates of potentially hazardous prescribing and inadequate blood-test monitoring

Fig 1 presents the primary outcomes of potentially hazardous prescribing and inadequate monitoring, relative to the intervention start date in each practice. Table B in S1 Table shows the rates, numerators, and denominators for each of the prescribing and monitoring outcomes

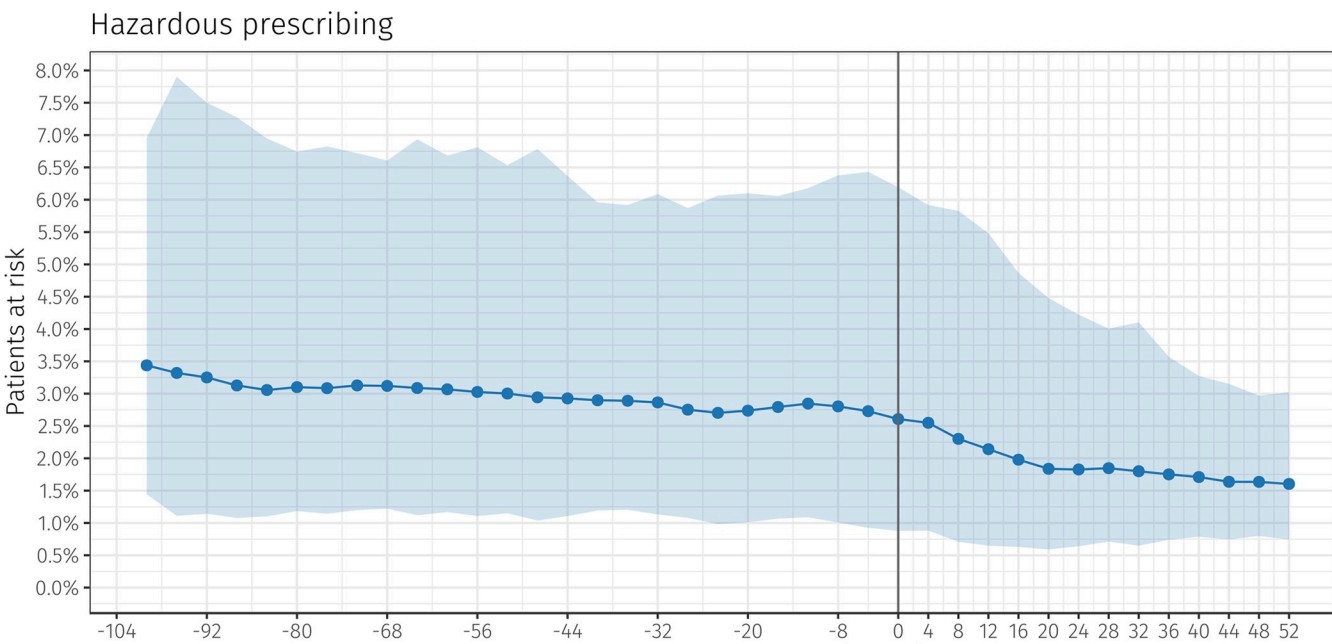

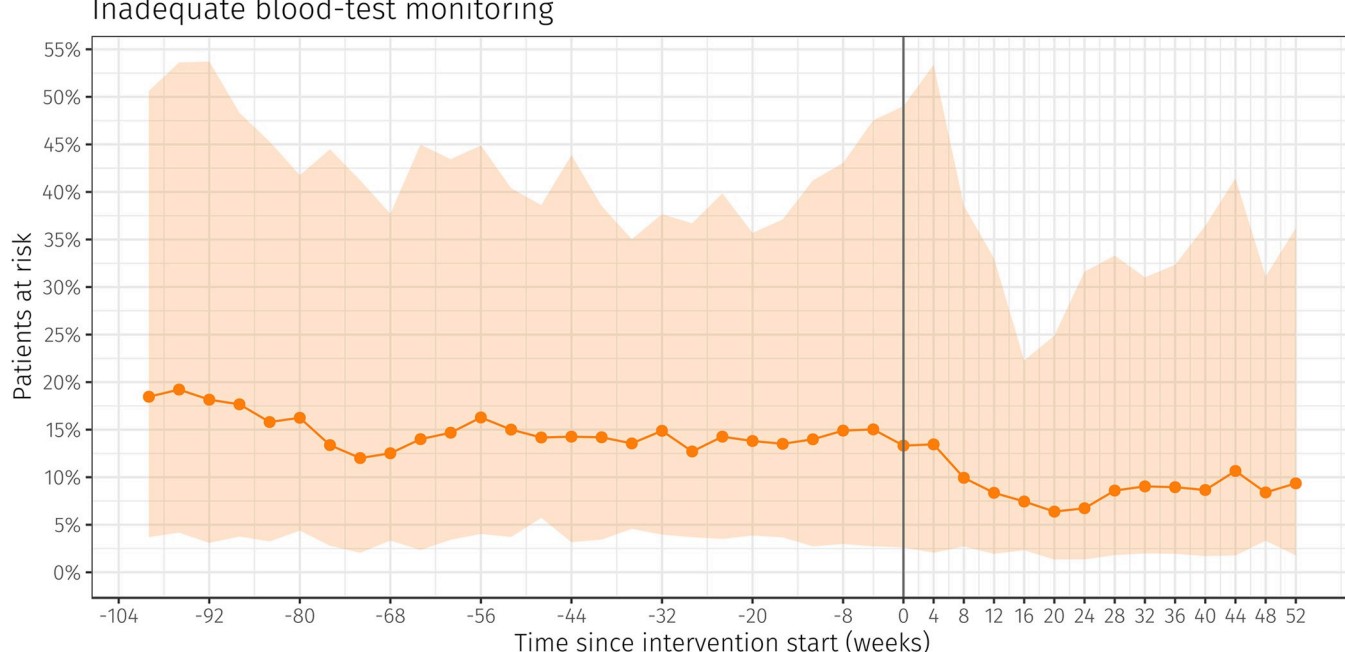

**Fig 1. Observed rates of patients exposed to medication safety hazards in 43 participating practices in Salford.** The outcomes were analysed every 4 weeks (indicated by the dots). The scales of the *y*-axes differ for the 2 outcomes. Inadequate blood-test monitoring was considered resolved when a new lab result was received in the electronic health record—typically several weeks after it was ordered. Shaded areas indicate 95% confidence intervals across practices.

at the different follow-up time points. At the start of the intervention, 2.61% (1,230 out of 47,183) of potentially exposed patients had received 1 or more potentially hazardous prescriptions; 13.32% (89 out of 668) had received inadequate blood-test monitoring. At this point in time, 95% of practices had rates of high-risk prescribing between 0.88% and 6.19% and rates of inadequate monitoring between 2.61% and 49.02% (shaded areas in Fig 1). At 12 months after the intervention start, the observed rate of hazardous prescribing had reduced by 1.01 percentage points to 1.60% (95% CI 1.49% to 1.72%; 657 out of 47,163 patients), and inadequate blood-test monitoring by 3.96 percentage points to 9.36% (95% CI 7.23% to 11.86%; 61 out of 652 patients). In terms of variation between practices at 12 months follow-up, 95% of practices had rates of high-risk prescribing between 0.74% and 3.02%, and rates of inadequate monitoring between 1.78% and 36.19%.

Table 2 shows for all indicators the rates at baseline and follow-up time points. In the analysis of the primary outcomes, the rate of potentially hazardous prescribing reduced faster after introduction of the SMASH intervention than would be expected from extrapolation of the pre-intervention trend, with an absolute difference at 4 weeks of −0.36 (95% CI −0.48 to −0.23, $p < 0.001$); 12 weeks, −0.50 (95% CI −0.60 to −0.39, $p < 0.001$); 24 weeks, −0.69 (95% CI −0.80 to −0.57, $p < 0.001$); and 12 months, −0.96 (95% CI −1.12 to −0.79, $p < 0.001$), corresponding to relative reductions of 14.1% (95% CI 8.0% to 21.4%), 19.9% (95% CI 13.7% to 27.2%), 27.9% (95% CI 20.3% to 36.8%), and 40.7% (95% CI 29.1% to 54.2%), respectively. The observed rate of inadequate blood-test monitoring also reduced faster than the extrapolated pre-intervention trend, with an absolute difference at 4 weeks of −2.40 (95% CI −4.53 to 0.08, $p = 0.058$); 12 weeks, −2.59 (95% CI −4.70 to −0.13, $p = 0.038$); and 24 weeks, −2.82 (95% CI −5.15 to −0.18, $p = 0.046$), corresponding to relative reductions of 17.8% (95% CI −0.5% to 41.4%), 19.6% (95% CI 0.8% to 44.3%), and 22.0% (95% CI 0.2% to 50.7%), respectively. At 12 months this difference was no longer significant (absolute reduction −2.85 [95% CI −5.68 to 0.71]; relative reduction 23.5% [95% CI −4.5% to 61.6%], $p = 0.127$).

Incident cases (i.e., arisen within the previous 30 days) of both hazardous prescribing (baseline rate, 0.42% [95% CI 0.36% to 0.48%]) and inadequate blood-test monitoring (baseline rate, 2.68% [95% CI 1.60% to 4.23%]) were lower than extrapolations of the pre-intervention trend up to 12 weeks after the intervention start, but not after 24 weeks or 12 months (Table 2).

Sustained reductions at 12 months were found for 5 types of hazardous prescribing, namely, oral NSAID without co-prescription of an ulcer-healing drug in patients aged ≥65 years (baseline rate, 2.28% [95% CI 2.08% to 2.49%]; 12-month change, −0.63 [95% CI −0.98 to −0.21], $p = 0.003$), antiplatelet drug without co-prescription of an ulcer-healing drug to patients with a history of peptic ulceration (baseline rate, 7.89% [95% CI 6.53% to 9.84%]; 12-month change, −2.62 [95% CI −3.23 to −1.95], $p < 0.001$), warfarin or NOAC in combination with an antiplatelet drug without co-prescription of an ulcer-healing drug (baseline rate, 3.44% [95% CI 2.70% to 4.31%]; 12-month change, −1.17 [95% CI −1.76 to −0.42], $p = 0.001$), aspirin in combination with another antiplatelet drug without co-prescription of an ulcer-healing drug (baseline rate, 3.33% [95% CI 2.82% to 3.91%]; 12-month change, −1.30 [95% CI −1.81 to −0.62], $p < 0.001$), and oral NSAID to patients with poor renal function (baseline rate, 1.57% [95% CI 1.08% to 2.19%]; 12-month change, −0.39 [95% CI −0.57 to −0.19], $p < 0.001$). The intervention was associated with changes in prescriptions of warfarin or NOAC in combination with an oral NSAID and of oral NSAID to patients with heart failure at, respectively, 12 and 24 weeks, but these changes were not sustained thereafter. For the latter type of hazardous prescribing, the average rate across all practices was still decreased at 12 months, but this decrease was no longer significant due to higher variation between practices (Table 2).

**Table 2. Effect of the SMASH intervention on primary and secondary outcomes relating to prescribing and monitoring safety.**

| Outcome measure | Number of patients exposed to (numerator) and having risk factors for (denominator) prescribing and monitoring hazards at baseline | Baseline percentage of hazardous prescribing and monitoring (95% CI) | Absolute difference in percentage (95% CI) of medication safety hazards at 4 weeks, 12 weeks, 24 weeks, and 12 months after intervention start as compared to extrapolations of the pre-intervention trends | | | |
|---|---|---|---|---|---|---|
| | | | 4 weeks after start | 12 weeks after start | 24 weeks after start | 12 months after start |
| Any prescribing hazard composite (1–10) | 1,230/47,183 | 2.61 (2.47 to 2.75) | **−0.36 (−0.48 to −0.23)** | **−0.50 (−0.60 to −0.39)** | **−0.69 (−0.80 to −0.57)** | **−0.96 (−1.12 to −0.79)** |
| Ongoing prescribing hazards | 1,032/47,183 | 2.19 (2.06 to 2.32) | **−0.31 (−0.43 to −0.19)** | **−0.47 (−0.56 to −0.36)** | **−0.67 (−0.77 to −0.55)** | **−0.96 (−1.11 to −0.79)** |
| New prescribing hazards | 198/47,183 | 0.42 (0.36 to 0.48) | **−0.05 (−0.09 to −0.01)** | **−0.04 (−0.08 to −0.00)** | −0.03 (−0.06 to 0.01) | −0.01 (−0.05 to 0.07) |
| 1. Prescription of an oral NSAID without co-prescription of an ulcer-healing drug in a patient aged ≥65 years | 472/20,746 | 2.28 (2.08 to 2.49) | −0.17 (−0.33 to 0.00) | **−0.31 (−0.47 to −0.13)** | **−0.47 (−0.67 to −0.23)** | **−0.63 (−0.98 to −0.21)** |
| 2. Prescription of an oral NSAID without co-prescription of an ulcer-healing drug to a patient with a history of peptic ulceration | 18/1,407 | 1.28 (0.76 to 2.01) | 0.05 (−0.05 to 0.16) | 0.06 (−0.03 to 0.16) | 0.08 (−0.01 to 0.19) | 0.11 (−0.05 to 0.28) |
| 3. Prescription of an antiplatelet drug without co-prescription of an ulcer-healing drug to a patient with a history of peptic ulceration | 111/1,407 | 7.89 (6.53 to 9.42) | **−0.47 (−0.89 to −0.03)** | **−0.79 (−1.20 to −0.36)** | **−1.45 (−1.91 to −0.96)** | **−2.62 (−3.23 to −1.95)** |
| 4. Prescription of warfarin or NOAC in combination with an oral NSAID | 44/3,545 | 1.24 (0.90 to 1.66) | **−0.13 (−0.25 to −0.00)** | **−0.14 (−0.25 to −0.02)** | −0.13 (−0.24 to 0.01) | −0.08 (−0.26 to 0.12) |
| 5. Prescription of warfarin or NOAC in combination with an antiplatelet drug without co-prescription of an ulcer-healing drug | 72/2,096 | 3.44 (2.70 to 4.31) | −0.50 (−0.95 to 0.03) | **−0.64 (−1.09 to −0.11)** | **−0.83 (−1.30 to −0.27)** | **−1.17 (−1.76 to −0.42)** |
| 6. Prescription of aspirin in combination with another antiplatelet drug without co-prescription of an ulcer-healing drug | 144/4,319 | 3.33 (2.82 to 3.91) | **−0.69 (−1.00 to −0.33)** | **−0.81 (−1.12 to −0.45)** | **−0.99 (−1.36 to −0.55)** | **−1.30 (−1.81 to −0.62)** |
| 7. Prescription of a non-selective beta-blocker to a patient with asthma | 314/23,276 | 1.35 (1.21 to 1.51) | −0.03 (−0.12 to 0.07) | −0.09 (−0.19 to 0.02) | −0.16 (−0.32 to 0.02) | −0.29 (−0.58 to 0.10) |
| 8. Prescription of a long-acting beta-2 inhaler (excluding combination products with inhaled corticosteroid) to a patient with asthma who is not also prescribed an inhaled corticosteroid | 38/277 | 13.72 (9.89 to 18.34) | −0.52 (−1.71 to 0.75) | −0.96 (−2.11 to 0.27) | −1.39 (−2.72 to 0.06) | −2.35 (−4.38 to 0.01) |
| 9. Prescription of an oral NSAID to a patient with heart failure | 37/2,523 | 1.47 (1.03 to 2.02) | **−0.21 (−0.37 to −0.06)** | **−0.19 (−0.32 to −0.04)** | **−0.21 (−0.37 to −0.03)** | −0.22 (−0.47 to 0.09) |
| 10. Prescription of an oral NSAID to a patient with chronic renal failure (eGFR < 45 ml/min/1.73 m$^2$) | 33/2,107 | 1.57 (1.08 to 2.19) | **−0.27 (−0.38 to −0.16)** | **−0.28 (−0.40 to −0.16)** | **−0.32 (−0.45 to −0.18)** | **−0.39 (−0.57 to −0.19)** |
| Any monitoring hazard composite (11–12) | 89/668 | 13.32 (10.84 to 16.14) | −2.40 (−4.53 to 0.08) | **−2.59 (−4.70 to −0.13)** | **−2.82 (−5.15 to −0.04)** | −2.85 (−5.68 to 0.71) |
| Ongoing monitoring hazards | 71/668 | 10.63 (8.39 to 13.22) | −0.99 (−2.50 to 0.81) | −1.32 (−2.86 to 0.53) | −1.76 (−3.43 to 0.30) | −2.16 (−3.99 to 0.21) |

(*Continued*)

**Table 2.** (Continued)

| Outcome measure | Number of patients exposed to (numerator) and having risk factors for (denominator) prescribing and monitoring hazards at baseline | Baseline percentage of hazardous prescribing and monitoring (95% CI) | Absolute difference in percentage (95% CI) of medication safety hazards at 4 weeks, 12 weeks, 24 weeks, and 12 months after intervention start as compared to extrapolations of the pre-intervention trends | | | |
|---|---|---|---|---|---|---|
| | | | 4 weeks after start | 12 weeks after start | 24 weeks after start | 12 months after start |
| New monitoring hazards | 18/668 | 2.68 (1.60 to 4.23) | **−0.74 (−1.18 to −2.70)** | **−0.66 (−1.14 to −1.13)** | −0.56 (−1.14 to 0.10) | −0.37 (−1.21 to 0.63) |
| 11. Prescription of methotrexate without both a recent full blood count and a liver function test in the last 3 months | 37/466 | 7.94 (5.65 to 10.78) | 0.27 (−0.76 to 1.41) | 0.17 (−0.76 to 1.19) | 0.00 (−1.00 to 1.12) | −0.31 (−1.80 to 1.44) |
| 12. Prescription of amiodarone for at least 6 months without a thyroid function test within the last 6 months | 52/203 | 25.62 (19.76 to 32.20) | −4.90 (−10.45 to 1.89) | −5.27 (−10.71 to 1.40) | −5.92 (−11.84 to 1.56) | −5.97 (−13.69 to 4.69) |

Bold font indicates significant difference ($p < 0.05$).

eGRF, estimated glomerular filtration rate; NOAC, non–vitamin K antagonist oral anticoagulant; NSAID, nonsteroidal anti-inflammatory drug; SMASH, Safety Medication dASHboard.

For both outcomes, pre-intervention levels, pre-intervention trends, and changes at follow-up varied substantially between practices. For potentially hazardous prescribing, the 12-month change ranged from −4.34 to +0.93 ($I^2$ = 91.1%; Fig G in S1 Fig). In 30 practices there was a decrease in potentially hazardous prescribing after 12 months; in 12 practices there was no statistically significant change; and in 1 practice there was an increase. For inadequate blood-test monitoring, the 12-month change ranged from −94.9 to +42.4 ($I^2$ = 83.4%; Fig J in S1 Fig). In several practices (e.g., practices 38 and 39) there was a marked increase in inadequate monitoring before the intervention and a sharp decrease afterwards, leading to high levels of estimated change. These 2 practices and 4 others achieved a significant reduction at 12 months, while in 29 practices there was no significant change, and 8 practices saw a (usually modest) increase. For both outcomes there was a negative correlation between the pre- and post-intervention trends (potentially hazardous prescribing, Pearson's correlation −0.353, $p = 0.020$; inadequate blood-test monitoring, Pearson's correlation −0.558, $p < 0.001$).

## Discussion

We evaluated a complex intervention to improve the safety of medicine prescribing and blood-test monitoring in general practice, combining training of general-practice-based clinical pharmacists, a web-based dashboard providing actionable feedback, and pharmacists reviewing individual patients' prescriptions and monitoring to initiate remedial actions or advise GPs on doing so. The intervention was associated with substantial and sustained reductions in targeted high-risk prescribing over 12 months after the start of the intervention, and marked reduction in the variation in rates of potentially hazardous prescribing between practices.

A gradual decrease was observed in potentially hazardous prescribing and inadequate monitoring during the first 20 weeks after initiating the intervention. For targeted high-risk prescribing, the decrease continued throughout the 12-month follow-up period, whereas for inadequate blood-test monitoring there was a decrease at 24 weeks, but not so at 12 months. Not surprisingly, we observed larger reductions in ongoing high-risk prescribing than in new high-risk prescribing, given that the intervention prompted review of patients who were already exposed to high-risk prescribing, although both rates were reduced.

We observed a small downward trend in both potentially hazardous prescribing and inadequate blood-test monitoring during the 2 years before the intervention (as shown in Fig 1). This was likely due to increasing awareness among some GPs of the risks associated with commonly prescribed medications targeted by our indicators, but there was substantial variation between practices, highlighting the potential for further improvements in medication safety through targeted practice-level intervention. Notably, the largest reductions were observed in practices with high prior rates of high-risk prescribing, and there was a marked reduction in the variation between practices during the 12-month follow-up period.

There was also marked variation in reductions of different forms of potentially hazardous prescribing. The largest reductions were observed in patients receiving warfarin or NOACs in combination with an antiplatelet drug without gastro-protective drug co-prescription, patients receiving aspirin with another antiplatelet drug; and patients with a history of peptic ulceration receiving antiplatelet drugs. All these forms of potentially hazardous prescribing can be easily resolved by discontinuing 1 or more of the medications if they are no longer clinically indicated or, alternatively, if appropriate, through co-prescribing of gastro-protective medication. In contrast, we did not observe a significant reduction in prescriptions of non-selective beta-blockers to patients with a history of asthma, despite evidence of increased risks of asthma exacerbation [37,38]. A partial explanation may be that GPs are hesitant to halt prescribing started by other specialist practitioners (e.g., psychiatrists or optometrists). Also, there exist instances where no suitable alternative treatment is available for the patient in question [38].

Interviews with healthcare professionals involved in the study [25] and analyses of activity logs of the dashboard [39] have indicated that in many practices, pharmacists were the primary (and sometimes, sole) users throughout the study. Activity logs showed that pharmacists, after an intensive review of patients identified on the dashboard for a given practice, switched to 'surveillance mode' for that practice, while focusing most of their efforts on new practices that started the intervention [39]. This surveillance was particularly noticeable for cases of inadequate monitoring which, by definition, started to be re-flagged on the dashboard after 6 months for those patients receiving long-term treatments, such as low dose methotrexate.

## Strengths and limitations of this study

To our knowledge no previous studies have evaluated changes associated with complex interventions that include ongoing, patient-level feedback on safe prescribing and monitoring in general practice. Our intervention used state-of-the-art web technology to provide feedback to practitioners using textual, numerical, and graphical information, benchmarking their practice's performance against the average performance for all participating practices and providing the ability to monitor their own performance over time [14]. This information was updated daily during the 12-month intervention period. At the same time, there was continued support of pharmacists to review high-risk cases and to initiate or advise on remedial actions.

The main strengths of our study are the pragmatic design, the range of clinically relevant outcomes evaluated, and the large number of general practices that took part. Forty-three out of 44 general practices in Salford took part in our study, eliminating potential forms of bias due to selection (e.g., volunteering). The high adoption of SMASH might have been partially driven by a small financial incentive. However, similar incentives have been included in the latest General Medical Services contract framework for general practices in the UK [40].

Our study had the following limitations. We evaluated the intervention in a pragmatic fashion during its rollout, and therefore did not randomise practices to receiving or not receiving the intervention. Instead, we enrolled practices in a 'naturalistic fashion' starting on different start dates; for logistical reasons (allocation of clinical pharmacists to practices), these dates

could not be randomised. Due to the lack of randomisation, we cannot exclude that unmeasured confounding has influenced our findings. However, the ITS design is recognised as one of the strongest quasi-experimental designs for assessing the impact of interventions in such circumstances [31]. The main threat to validity is bias through secular trends in outcomes caused by other interventions (such as new policies). While no new relevant policies came into force during our study, there was an increasing awareness of the importance of medication safety in primary care in the UK during this period. For this reason, we applied a 2-stage statistical analysis strategy, comparing model extrapolations of pre-intervention trends at the practice level to observed outcomes post-intervention. Because practices started the intervention over a long time period (April 2016 to September 2017), we believe that the risks of bias are low. We do acknowledge that half of the participating practices started the intervention during the last 2 months of 2016 (Table 1), and that there were some differences between this group and the practices that started earlier or later. Furthermore, all our outcomes were measured by interrogating routinely collected electronic health records. It is possible that medication safety risks were not detected due to omissions or mistakes in patients' records. It is also possible that some apparent instances of hazardous prescribing or inadequate blood-test monitoring were caused by mistakes in patients' records and were resolved by correcting those mistakes rather than through clinical actions.

## Comparison with other studies

Several feedback-based, complex interventions targeting medication safety in primary care have been evaluated in recent years, the results of which are consistent with our findings. Avery et al. [12] randomised 72 general practices in England to simple feedback (lists of at-risk patients plus brief written educational materials) accompanied by pharmacist-led educational outreach meetings (PINCER) versus simple feedback alone. Feedback, consisting of a paper report, was repeated after 6 and 12 months, but pharmacist support was limited to 12 weeks. The effects on high-risk prescribing were similar to changes found in our study at 24 weeks but waned at 12 months. It was already known that feedback is more effective when it is provided more than once [13], but this result shows that when feedback is part of a larger intervention, the entire intervention must be provided more than once—not just the feedback. In our case, the continued support from practice-based pharmacists and ongoing access to up-to-date information through the dashboard made a crucial difference to sustainably target high-risk prescribing in the highly committed general practice environment.

In a cluster-randomised, stepped-wedge trial conducted in 33 practices, Dreischulte et al. [41] evaluated a complex intervention comprising electronic feedback similar to SMASH, 1-hour pharmacist outreach visits and financial incentives. They observed a reduction in high-risk prescribing after 12 months similar to what we found in our study. Building on this, MacBride-Stewart et al. [42] evaluated the impact of a safety improvement initiative comprising an educational workshop, a single round of benchmark feedback, financial incentives, and electronic search tools to identify high-risk patients. There were substantial and sustained reductions in the high-risk prescribing of NSAIDs, although with some waning of effect 12 months after the intervention ceased. The same intervention had no effect on antipsychotic prescribing in people with dementia.

## Policy implications

Emerging evidence suggests that general-practice-based pharmacists enhance integrated patient care and can deliver clinical interventions efficiently and in high volume [43,44]. Our study demonstrates how these pharmacists can also play a key role in sustainably improving

medication safety in general practice and reducing variation in potentially hazardous prescribing between practices. However, for this to be effective, it is imperative to have an underpinning 'learning health systems' [45] capability for continuous data-driven self-study that promotes change and improvement. Specifically, it requires a digital infrastructure for continuous, patient-level feedback that allows practitioners to identify high-risk patients, access educational material, and compare rates of high-risk prescribing and monitoring both between practices and within their practice over time.

Our intervention easily scales up because it is based on the automated analysis of routine healthcare data, and can therefore be used in any environment that uses electronic prescribing. As such it can be used to support the WHO's Third Global Patient Safety Challenge [1], which focuses on reducing medication-related iatrogenic harm. Specifically, it can be used to target high-risk situations and constitutes a programme of action for designing safe systems and practices for medication in primary care.

## Conclusion

Our intervention was associated with reduced rates of potentially hazardous prescribing and inadequate blood-test monitoring in general practices during the first 24 weeks after initiation. This reduction was sustained at 12 months for prescribing but not for monitoring. There were substantial differences in rates of potentially hazardous prescribing between practices prior to the intervention, and there was a marked reduction in the variation between practices over the 12-month follow-up period.

## Supporting information

**S1 Fig. Supplemental figures.** Screenshots from the SMASH interactive dashboard (Figs A–D) and forest plots showing variation between practices in the reduction in potentially hazardous prescribing and inadequate monitoring (Figs E–J).
(DOCX)

**S1 Study Protocol. Protocol for the study.**
(PDF)

**S1 Tables. Supplemental tables.** Table A. Definitions of the medication safety indicators targeted by the SMASH intervention. Table B. Rates of hazardous prescribing and inadequate medication monitoring for 43 general practices in Salford.
(ZIP)

**S1 TIDieR Checklist. Completed TIDieR Checklist.**
(PDF)

**S1 TREND Statement. Completed TREND statement.**
(PDF)

## Acknowledgments

The work uses data provided by patients and collected by the NHS as part of their care and support. PRIMIS at the University of Nottingham has provided detailed specifications for the safety indicators used in SMASH. We are grateful to the NHS Salford Clinical Commissioning Group and to all the pharmacists and all other staff at the general practices for taking part in the study.

## Author Contributions

**Conceptualization:** Niels Peek, Richard N. Keers, Richard Williams, Mark Jeffries, Denham L. Phipps, Benjamin Brown, Anthony J. Avery, Darren M. Ashcroft.

**Data curation:** Wouter T. Gude, Richard N. Keers, Richard Williams, Evangelos Kontopantelis.

**Formal analysis:** Evangelos Kontopantelis.

**Funding acquisition:** Anthony J. Avery, Darren M. Ashcroft.

**Investigation:** Niels Peek, Richard N. Keers, Richard Williams, Mark Jeffries, Denham L. Phipps, Darren M. Ashcroft.

**Methodology:** Niels Peek, Wouter T. Gude, Evangelos Kontopantelis, Benjamin Brown, Anthony J. Avery, Darren M. Ashcroft.

**Resources:** Niels Peek.

**Software:** Richard Williams.

**Supervision:** Niels Peek, Anthony J. Avery, Darren M. Ashcroft.

**Writing – original draft:** Niels Peek, Wouter T. Gude, Darren M. Ashcroft.

**Writing – review & editing:** Niels Peek, Wouter T. Gude, Richard N. Keers, Richard Williams, Evangelos Kontopantelis, Mark Jeffries, Denham L. Phipps, Benjamin Brown, Anthony J. Avery, Darren M. Ashcroft.

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
