## [Decision Letter · Decision Letter 0]

9 Mar 2020

Dear Dr. Peek,

Thank you very much for submitting your manuscript "Impact of a pharmacist-led actionable audit and feedback intervention to improve medication safety in primary care: interrupted time series analysis" (PMEDICINE-D-19-04218) for consideration at PLOS Medicine. 

Your paper was discussed among the editorial team and sent to independent reviewers, including a statistical reviewer. The reviews are appended at the bottom of this email and any accompanying reviewer attachments can be seen via the link below:

[LINK]

In light of these reviews, we will not be able to accept the manuscript for publication in the journal in its current form, but we would like to invite you to submit a revised version that fully addresses the reviewers' and editors' comments. You will appreciate that we cannot make a decision about publication until we have seen the revised manuscript and your response, and we expect to seek re-review by one or more of the reviewers. 

We hope to receive your revised manuscript by Mar 30 2020 11:59PM. Please email us (plosmedicine@plos.org) if you have any questions or concerns.

Please let me know if you have any questions. Otherwise, we look forward to receiving your revised manuscript in due course. 

Sincerely,

Richard Turner PhD, for Caitlin Moyer, Ph.D.

Associate Editor, PLOS Medicine

rturner@plos.org

We note that you are unable share data publicly (even when de-identified, we assume). Are you able to include aggregate data in supplementary files for the benefit of other researchers, or via a publicly-accessible repository?

Noting the research design, we ask you to adopt softer wording in places where you currently assert that "the SMASH intervention reduced rates ..." and similar. For example, "Implementation of the SMASH intervention was associated with reduced rates of ..." would seem more apt. 

Please substitute "this reduction" or "this apparent impact" for "this effect". 

We suggest an amended title: "Evaluation of a pharmacist-led audit and feedback intervention for improving medication safety in primary care: an interrupted time-series analysis". 

Please reformat your abstract to comprise three subsections: "background"; "methods and findings"; and "conclusions". The final sentence of the "methods and findings" subsection should quote 2-3 of the study's main limitations. 

To your abstract, please add aggregate demographic details for study participants; and study dates. 

In your abstract and throughout the paper, please quote p values alongside 95% CI, where available. 

After the abstract, we will need to ask you to add a new and accessible "author summary" section in non-identical prose. You may find it helpful to take a look at one or two recent research papers published in PLOS Medicine to get a sense of the preferred style. 

Early in the methods section, please state whether the study had a protocol or prespecified analysis plan, and if so add the relevant document(s) as a supplementary file, referred to in the text. Please highlight analyses that were not prespecified. 

We note that you mention the TREND statement in the methods section. Please add a completed checklist for the most appropriate reporting guideline, which may well be TREND, as a supplementary document, referred to in your methods section. In the checklist, please refer to individual items by section (e.g., "Methods") and paragraph number rather than by page or line numbers, as the latter generally change in the event of publication. 

Please avoid claims of "the first" and the like, noting one instance in your discussion section, and where necessary please add "to our knowledge" or similar. 

We ask you to revisit the "strengths and limitations" section, aiming to correct an apparent imbalance in favour of the former component. For example, where you note the absence of randomization, the possible influence of unmeasured confounders could be quoted. 

Where you mention the word "trend", please state the statistical basis for this; or substitute a different word or phrase, e.g. "apparent reduction". 

Please revisit your reference list to ensure that all citations meet journal style. All italics and boldface text should be converted to plain text; and where appropriate, 6 authors names should be listed, followed by "et al.".

Please ensure that all references have full access details, noting reference 20, for example. 

Comments from the reviewers:

*** Reviewer #1: 

This is a statistical review of manuscript PMEDICINE-D-19-04218. The manuscript reports the results of a pharmacist-led intervention of medication safety in primary care. 

My comments are here below. 

* Abstract - Results: I suggest that you start by providing the rates at baseline ("At baseline, 95% of practices had rates…"), before presenting your results. This will allow the reader to understand the starting point, and realise what the percentage reductions that are mentioned are relative to in a more "chronological" way. 

* Statistical analysis: Could you please provide more information on your sample size calculation? Please summarise in a few words the reference levels for which you use reference 6. Also please clarify what you mean by a before-after study? Is it effectively a change in percentage? What is your estimated population size of a GP? Finally, 88.3% power is slightly unusual. Could you please clarify?

* Statistical analysis: the sample size calculation is 30 GPs. But you include 43 GPs. Please could you clarify why? 

* Results: you mention that "the rate of [] prescribing reduced significantly more after introduction of the SMASH intervention compared to extrapolations". Does this mean that you compare the projected rate of 12 months, using data from the pre-intervention period, to the observed rate at 12 months? Or effectively is it the case that you are comparing the observed rates to the rates at baseline? 

*** Reviewer #2: 

[see attachment]

*** Reviewer #3: 

Reviewers Comments 

Impact of a pharmacist-led actionable audit and feedback intervention to improve medication safety in primary care: interrupted time series analysis

This paper reports the outcomes of an intervention in an NHS England primary care setting (GP practices in part of Greater Manchester). 

The intervention was described as having 3 components

1. the placement of clinical pharmacists into GP practices who are trained to identify and resolve root causes of medication errors,

2. the feedback of potentially hazardous prescribing/blood monitoring via an interactive dashboard that provides patient-level analysis, patient details and daily updates in 12 measures of potential harm (10 prescribing, 2 blood monitoring) selected from PINCER trial measures, and

3. a review of patient records and ordering of blood tests or changing to prescribing (or recommendation to prescriber to change) where appropriate to specifically address the observed potentially hazardous prescribing/blood monitoring. 

In the majority of cases the review was undertaken by the clinical pharmacist and only one of these was an independent prescriber so most changes to prescribing were recommended to a GP within the practice. 

The effect of the intervention was estimated by linear regression of interrupted series data (log-transformed to adjust for curves resulting from ceiling and floor effects). Pre-intervention trends, step changes and post-intervention trends were used to estimate the expected and observed values and, from these measurements, the effect size at 4, 12, 24 and 52 weeks post-intervention were estimated. These were estimated for the proportion of patients exposed to a composite of the 10 prescribing and separately the 2 blood monitoring measures. Secondary outcomes were the estimated effect size of the intervention for 1. the proportion of patients with long-standing potentially hazardous prescribing/blood monitoring and 2. for the proportion of patients newly exposure to the potentially hazardous prescribing/blood monitoring. 

Additional reported analysis were 

- the estimated effect size at 4, 12, 24 and 52 weeks was reported for each of the individual indicators. 

- the estimated effect size at 52 weeks for each of the two composite measures (prescribing, blood monitoring) was separately analysed then meta-analysed and forest plotted with I2 estimated.

- the rates for the composite potentially hazardous prescribing/blood monitoring (4 weekly intervals for the 2 years prior and 1 year post intervention) are plotted with 95% confidence intervals for all practices. 

The outcome was a reduction in the composite measure of potentially hazardous prescribing at 4, 12, 24 and 52 weeks. Reductions in composite measure of blood monitoring were observed at 12, 24 weeks but not at 4 or 52 weeks. 

The other reported results show that the intervention had an effect on some but not all potentially hazardous prescribing measures and for neither of the individual blood monitoring measures. 

The authors conclude that the intervention has reduced potentially hazardous prescribing which is sustained over time. They highlighted the characteristics of the intervention that are different from others which help to explain why the effect is sustained. 

Major issues 

1.

The major issue with the evaluation is stating that the intervention resulted in a change when the confidence intervals of the estimated effect size cross zero. For example in the abstract (page 8); results section 2nd sentence is the following statement:

The rate of inadequate blood-test monitoring reduced by 22.0% (95% CI, 0.2% to 50.7%) at six months and by 23.5% (95% CI, -4.5% to 61.6%) at 12 months.

Given the confidence interval of the 12 months figure crosses zero the reduction is not statistically significant (at a 0.05 significance level). 

This is repeated in the body of the paper in the Rates of potentially hazardous prescribing and inadequate blood-test monitoring section, on page 15. 

The rate of inadequate blood-test monitoring reduced significantly more compared to the extrapolated pre-intervention trend with an absolute difference at four weeks of -2.40 (95% CI, -4.53 to 0.08), twelve weeks -2.594 (95% CI, -4.70 to -0.13), 24 weeks -2.824 (-5.15 to -0.18), and 52 weeks -2.85 (95% CI, -5.68 to 0.71), corresponding to relative reductions of 17.8% (95% CI, 0.5 to 41.4), 19.6% (95% CI, 0.8 to 44.3), 22.0% (95% CI, 0.2 to 50.7%), and 23.5% (95% CI, -4.5 to 61.6) respectively.

Again the confidence intervals for the absolute rates cross zero in the 4 week and 52 week figures so there is no statistically significant reduction (at a 0.05 significance level) for these two times. It would be helpful if the authors could explain how the relative reduction at 4 weeks is statistically significant when the absolute at 4 weeks is not.

In light of these comments the statements in the conclusion would need to be reviewed as there is no statistically significant reduction in the blood monitoring (95% confidence) at 12 months following the implementation of the intervention.

2.

The methods describe describes the practice-level analysis being done at 4, 12, 24 and 52 weeks but only the 52 week results are presented (in the last paragraph of the results section and as the supplementary figures 2 and 3). Related to this, an interpretation of the weaker correlation between pre and post intervention trends in the composite measure of potentially hazardous prescribing compared to the composite measure of blood monitoring would be useful to make it clearer why this statistic is being reported. 

3.

In the discussion, there is a statement at the top of page 11 "In other words, our intervention appeared to not only accelerate the pre-intervention trend in practices where it existed but also initiated it where it was absent." No evidence or analysis that showed those practices with higher levels of potentially hazardous prescribing prior to the intervention did not have declining trends leading up to the intervention was provided. Was that analysis done but not provided/reported?

4. 

It might be useful to the reader if the authors were able to suggest reasons for the difference in outcome for the difference measures using the results from the qualitative analysis. Understanding why some measures are unchanged by this intervention (e.g. prescribing beta-blockers to patients with a diagnosis of asthma) would help others when they replicate this initiative as it might mean some types of potentially hazardous prescribing require a totally different type of intervention in order to have an effect or might be that an adjustment of this intervention is what is required. 

5. 

The 12 PINCER indicators were selected based on their prevalence nationally and in Salford; was it low prevalence or high prevalence (or some other criteria) used to select the 12. Readers might benefit from seeing the PINCER measures not selected and the prevalence of all of them. 

6.

The paper describing the evaluation of the intervention follows the TREND style. A more structured description of the intervention might help the reader assess whether the intervention could be replicated in their own setting e.g. the template for intervention description and replication (TIDieR). For example, replicating this in other areas it would be useful to know whether the medication reviews were done with or without the patient present. 

Minor issues

1. 

In Table 1 please provide an explanation for the p-value in the final column - what measures do the p-values relate to?

2.

The results were measured at 4, 12, 24 and 52 weeks but sometimes the 24 week and 52 week results are described as six month and 12 month (and once the 52 weeks is described as one year). It would help if the description of the timeframe of the post-intervention analysis was consistent and accurate. 

3.

Results in the text sometimes have 3 significant figures, 2 are used in table 2 and 1 in supplementary table 2. It would help if the used of significant figures was consistent. 

4. 

In the methods section bottom of page 6 the statement "All outcomes data were collected by interrogating primary care electronic health records" is this a separate extract/database to the SIR database? If it is separate would it be possible to provide additional details sufficient to allow replication by other researchers?

***

[LINK]

---

## [Decision Letter · Decision Letter 1]

2 Jul 2020

Dear Dr. Peek,

Thank you very much for re-submitting your manuscript "Evaluation of a pharmacist-led actionable audit and feedback intervention for improving medication safety in primary care: an interrupted time series analysis" (PMEDICINE-D-19-04218R1) for consideration at PLOS Medicine.

I have discussed the paper with editorial colleagues and it was also seen again by two reviewers. I am pleased to tell you that, provided the remaining editorial and production issues are dealt with, we expect to be able to accept the paper for publication in the journal.

[LINK]

Please let me know if you have any questions. Otherwise, we look forward to receiving the revised manuscript shortly. 

Sincerely,

Richard Turner, PhD

rturner@plos.org

Requests from Editors:

Please check the Dryad code for data access, which we were unable to access. 

Please adapt the title slightly to "... in UK primary care ...". As an alternative, you might wish to add "in Manchester, UK" to the segment of the title before the colon. 

Please make that "43" in the abstract.

Please adapt the end of the "Methods and findings" subsection of your abstract so that the final sentence begins "Study limitations include ..." or similar. The quoted limitations could include the fact that practices were not randomized, for example, and the possibility that unmeasured confounders influence your findings. 

At the end of p.4 of the PDF, please make that "Our objective was to evaluate".

On p.9 of the PDF, you present various "significant reductions", but we note that not all the point estimates are significant at the 5% level. Please modify the text as appropriate. 

Please signpost the discussion of limitations in the main text with "This study had some limitations ..." or similar. As in the abstract, you may wish to note the possible influence of unmeasured confounders on your findings. 

On p.12 of the main text, please make that "... it is also possible that some". 

Please remove the mention of funding at the end of the main text. This information will appear in the metadata upon publication, via information in the submission form. 

We ask you to review the full paper to ensure that the wording does not imply that causality has been established in this research design. For example, at the top of p.11 of the PDF there is the phrase "... no previous studies have evaluated the effects ...", in which "impact" could be substituted. The author summary section also includes similar phrasing, and there is similar wording at the end of p.9 and on p.10 of the PDF ("12-month effect").

Please update reference 38. 

The TREND and TIDieR attachments both contain page numbers, which we ask you to remove (as page numbers generally change upon publication). In the checklists, please refer to individual items by section (e.g., "Methods") and paragraph number. 

Please refer to the attachments in your methods section (e.g., "See S1_TREND_Checklist"). 

Comments from Reviewers:

*** Reviewer #1: 

This is a statistical review of manuscript PMEDICINE-D-19-04218_R1. The authors have satisfactorily replied to my previous comments. Thank you. I have no further points. 

*** Reviewer #3: 

Thank you for considering the reviewers comments and agreeing to make changes to the original submitted paper.

I'll limit my feedback to the comments I originally made as reviewer#3 and the changes made.

R3.1. The major issue with the evaluation is stating that the intervention resulted in a change when that change is statistically insignificant. I had thought non-statistically significant changes would not be even mentioned in the abstract or described in words and well as figure. In the abstract the way the following is written it could be misread and readers might think there was a reduction at 12 months where there wasn't - "The rate of inadequate blood-test monitoring (composite of 2 indicators) reduced by 22.0% (95% CI, 0.2% to 50.7%, p=0.046) at 24 weeks and by 23.5% (95% CI, -4.5% to 61.6%, p=0.127) at 12 months." The authors are clearer in the discussion when they write "inadequate blood-test monitoring there was a decrease at 24 weeks, but not so at 12 months."

R3.2. thank you for including all the forest plots

R3.4 thank you for adding some thoughts as to why the response for each safety measure differed 

R3.5. thank you for clarifying it was PINCER measures with a higher prevalence included in your feedback

R3.6. thank you for completing the TIDier statement

R3.7. thank you for removing the unnecessary p-values

R3.8. thank you for being consistent when describing the result times. "We agree and now use 4 weeks, 12 weeks, 24 weeks, and 12 months consistently." - it seems a little strange to choose to use a mix of weeks and months 

R3.9. thank you for consistently using the same number of significant figures

R3.10. thank you for clarifying the source of the extract used for the analysis 

These additional comments relate to changes made to the original paper:

R3.11. in the "What do these findings mean?" section at the start of the paper the authors state

"Electronic, actionable audit and feedback interventions can provide long-term improvements to medication safety in general practice - but only when they are delivered on an ongoing basis."

I am not sure the phrase "but only" can be justified from the findings of this study.

R3.12. In the introduction (top of page 4) the following phrase 

"means that medication errors in primary care contribute to more than seven times more avoidable deaths than secondary care (627 versus 85)" has the more in it twice making it difficult to read. Perhaps "means that avoidable deaths in primary care due to medication errors are seven times more likely than in secondary care (627 versus 85)" is clearer.

R3.13. Page 11 in discussion "after an intensive review of patents" should be "after an intensive review of patients"?

R3.14. Page 12 in discussion "The main thread to validity is bias" should be "The main threat to validity is bias"

***

[LINK]

---

## [Editor Report · Decision Letter 2]

8 Sep 2020

Dear Prof. Peek, 

On behalf of my colleagues and the academic editor, Dr. Sean MacBride-Stewart, I am delighted to inform you that your manuscript entitled "Evaluation of a pharmacist-led actionable audit and feedback intervention for improving medication safety in UK primary care: an interrupted time series analysis" (PMEDICINE-D-19-04218R2) has been accepted for publication in PLOS Medicine. 

PRODUCTION PROCESS

PRESS

PROFILE INFORMATION

Thank you again for submitting the manuscript to PLOS Medicine. We look forward to publishing it. 

Best wishes, 

Richard Turner, PhD

Senior Editor 

PLOS Medicine

plosmedicine.org